# Integrin α_E_(CD103)β_7_ in Epithelial Cancer

**DOI:** 10.3390/cancers13246211

**Published:** 2021-12-09

**Authors:** Johanna C. Hoffmann, Michael P. Schön

**Affiliations:** 1Department of Dermatology, Venereology and Allergology, University Medical Center Göttingen, 37075 Göttingen, Germany; johanna.hoffmann@med.uni-goettingen.de; 2Lower Saxony Institute of Occupational Dermatology, University Medical Center Göttingen, 37075 Göttingen, Germany

**Keywords:** CD103, integrin, α_E_(CD103)β_7_, E-cadherin, T_RM_ cells, immunotherapy, immunosurveillance, skin cancer, squamous cell carcinoma, basal cell carcinoma

## Abstract

**Simple Summary:**

The immune system in cancer is a central focus of research and clinical developments alike. We delineate our current view on α_E_(CD103)β_7_ integrin (CD103) expressing tumor infiltrating T lymphocytes in epithelial tumors. CD103 binds to E-cadherin within epithelial tissues and can be induced by TGFβ. It appears to play a role in the formation and function of the immunological synapse between cytotoxic T cell and tumor cell. Infiltration of CD103-expressing T cells in epithelial tumors is often associated with a better prognosis for patients. An exception seems to be epithelial skin tumors, where CD103 expression is not associated with better prognosis. We also show new data that CD103 is significantly more highly expressed in squamous cell carcinomas of the skin than in basal cell carcinomas or some other skin tumors, although the overall expression pattern is heterogeneous. A better understanding of α_E_(CD103)β_7_ integrin may facilitate some immunological antitumor therapies.

**Abstract:**

Interactions of both the innate and the adaptive immune system with tumors are complex and often influence courses and therapeutic treatments in unanticipated ways. Based on the concept that CD8^+^T cells can mediate important antitumor effects, several therapies now aim to amplify their specific activity. A subpopulation of CD8^+^ tissue-resident T lymphocytes that express the α_E_(CD103)β_7_ integrin has raised particular interest. This receptor presumably contributes to the recruitment and retention of tumor-infiltrating immune cells through interaction with its ligand, E-cadherin. It appears to have regulatory functions and is thought to be a component of some immunological synapses. In TGF-rich environments, the α_E_(CD103)β_7_/E-cadherin-interaction enhances the binding strength between tumor cells and infiltrating T lymphocytes. This activity facilitates the release of lytic granule contents and cytokines as well as further immune responses and the killing of target cells. Expression of α_E_(CD103)β_7_ in some tumors is associated with a rather favorable prognosis, perhaps with the notable exception of squamous cell carcinoma of the skin. Although epithelial skin tumors are by far the most common tumors of fair-skinned people, there have been very few studies on the distribution of α_E_(CD103)β_7_ expressing cells in these neoplasms. Given this background, we describe here that α_E_(CD103)β_7_ is scarcely present in basal cell carcinomas, but much more abundant in squamous cell carcinomas with heterogeneous distribution. Notwithstanding a substantial number of studies, the role of α_E_(CD103)β_7_ in the tumor context is still far from clear. Here, we summarize the essential current knowledge on α_E_(CD103)β_7_ and outline that it is worthwhile to further explore this intriguing receptor with regard to the pathophysiology, therapy, and prognosis of solid tumors.

## 1. A Brief History of the α_E_(CD103)β_7_ Integrin in Science

More than three decades of intensive research and about 2000 publications (MedLine, accessed on 7 October 2021) substantiate that the α_E_(CD103)β_7_ integrin (hereafter referred to as CD103) is an interesting and important player in numerous immunological reactions. Nevertheless, many aspects of its (patho)physiological significance remain enigmatic to this day. CD103 is a heterodimeric transmembrane surface receptor expressed by several cell types of the immune system. First discovered in the late 1980s as a marker for intestinal intraepithelial T cells in humans, mice, and rats [1,2], CD103 is thought to mediate lymphocyte retention in epithelial tissues through binding to E-cadherin [3,4]. It is found on a rather illustrious assortment of different cells including innate lymphoid cells [5], mast cells [6] as well as some intestinal, splenic, central nervous, and cutaneous dendritic cells [7,8]. Yet, CD103 is preferentially expressed by epithelial-associated T lymphocytes of the adaptive immune system, in particular some CD8^+^ T cells (CTLs and tissue-resident memory T cells (T_RM_)) as well as CD4^+^ regulatory T cells (T_reg_) [9,10,11]. T_RM_ are a subpopulation of cytotoxic T cells [12] that help to protect epithelia against infections, e.g., by viruses [12,13]. Besides the expression of CD103, the T_RM_ phenotype is characterized by the expression of VLA-1 (very late activation antigen, α_1_(CD49)β_1_(CD29) integrin), and the C-type lectin CD69 (very early activation antigen). This armamentarium arguably contributes to the tissue retention and function of T_RM_ [14]. Briefly, T_RM_ express a broad range of adhesion and chemokine receptors. Among these, CCR5 and CCR6 are thought to confer T-cell homing to the inflammatory tumor microenvironment [15]. The former is also involved in functions of the CD103-dependent immunological synapse between T cells and tumor target cells [16]. A role for CD103 in T cell homing to epithelia has been suggested previously [17]. Tissue-specific antigen expression patterns, differentiation, and maturation of TRM have been reviewed recently [18,19].

Solid epithelial tumors often show infiltration by T_RM_ cells expressing CD103. Hence, following the observation that CD103 expression may be linked to a more favorable prognosis in some tumors, clinical and scientific interest has recently focused on this receptor [20]. However, the role of CD103 or CD103-expressing cells in solid tumors is far from clear. Currently, much research is being carried out to clarify this issue [18]. It is more surprising that only very few data on CD103 are available on epithelial skin cancer, which is by far the most common malignant tumor in humans [19,20]. Here, we review the still quite enigmatic role of CD103 in epithelial tumors and contribute some of our own data on its expression in basal cell carcinomas and squamous cell carcinomas of the skin.

## 2. Structure and Function of CD103

The integrin α_E_ chain (CD103 in the strict sense) dimerizes exclusively with the β_7_ integrin subunit (Figure 1) while the latter can also pair with α_4_, resulting in the α_4_(CD49d) β_7_ receptor (lymphocyte Peyer’s patch adhesion molecule) [21,22]. Scientific interest in CD103 increased after it was discovered that it is the first integrin to bind to a cadherin, namely E-cadherin [3,4,23,24]. The α_E_-subunit binds through its metal ion-dependent coordination site (MIDAS) motif in domain 1 to the tip of the BC-loop within the EC1 domain of E-cadherin [23,24,25,26] (Figure 1). The best-known function of E-cadherin is certainly homophilic binding to identical E-cadherin molecules [27], although there is also (circumstantial) evidence for binding to other cadherins [28]. In the case of CD103, however, the binding is heterophilic. The β_7_ chain does not bind directly to E-cadherin but appears to have more of a regulatory function in this process [27]. Experimental evidence suggests that in some cases, for example in keratinocytes and endothelial cells, CD103 can also bind to other ligands that have not been identified until now [29,30]. In some cell types, cadherin26 (CDH26) has been suggested as a possible additional ligand [31]. Binding through CD103 allows immunocytes to maintain close contact with both peripheral lymphoid and non-lymphoid tissue, e.g., in the lung, skin, gastrointestinal, and genitourinary tracts [25].

Notwithstanding extensive attempts to identify factors that induce CD103, only transforming growth factor-β (TGFβ) has been found to consistently upregulate its expression, e.g., in human leukocyte cultures [32,33,34]. This cytokine induces CD103 on human CTLs in conjunction with T cell receptor (TCR) activation [35,36]. TGFβ induces expression of CD103 more strongly on CD8^+^ cells than CD4^+^ cells, thus suggesting preferential inducibility on CD8^+^ cells [34]. While TGFβ also conveys negative signals toward IL-1 and IL-2-dependent proliferation of activated T cells [37,38], B cells [39], and myeloid cells [40], it appears to play a key role for CD103-dependent immunity within tumor microenvironments [32,36].

## 3. CD103 and Epithelial Tumor Cells: A Multifaceted Relationship

CD103-expressing tumor-infiltrating leukocytes bind more strongly under shear flow to autologous tumors expressing E-cadherin [41]. However, E-cadherin can be downregulated in malignant tumors (for example, during epithelial-mesenchymal transition). In some cases, this downregulation has been linked to increased invasive or metastatic potential, although this association is by no means universal nor always straightforward [42,43,44,45,46]. In any case, it could be hypothesized that in cases of reduced E-cadherin expression, migration of CD103-expressing immune cells should also decrease. However, this plausible hypothesis has, so far, hardly been corroborated by actual data and there appears to be no direct correlation between E-cadherin levels and the presence of CD103-expressing immune cells within or around tumors, even though some studies suggest such an association in certain cases [47,48]. This, in turn, suggests that there may be other-arguably as yet undiscovered ligands for CD103. Likewise, it is conceivable that CD103 also has functions independent of ligand binding after immune cells are recruited intratumorally through other interactions.

Although the exact manner in which CD103 modulates leukocyte functions inside or around solid tumors is still not clear, several mechanisms that could contribute to this interaction can be delineated (schematically depicted in Figure 2). In general, it is likely that CD103 contributes to tumor-directed activities in concert with other relevant molecules, for example, CD39 (ecto-ATP diphosphohydrolase 1 expressed by some TILs) [49]. A tumor microenvironment rich in TGFβ induces phosphorylation of SMAD 2/3. The latter complexes with SMAD 4 and initiates the SMAD signaling pathway. The SMAD complex triggers the transcription of the *ITGAE* gene thus producing CD103 mRNA and, eventually, cell surface expression of CD103 [50]. Finally, cell–cell contact is established between the α_E_ subunit and the EC1-domain of E-cadherin.

In addition, TGFβ directly increases the affinity of CD103 for E-cadherin through phosphorylation of integrin-linked kinase (ILK) [51]. T cell receptor (TCR) signaling is stimulated via the nuclear factor of activated T cells (NFAT) pathway phospholipase C (PLC) γ1, inositol trisphosphate (IP3), and, depending on calcium and calcineurin, the phosphorylation of NFAT. The latter contributes to the transcription of the *ITGAE* gene and the production of CD103 mRNA [52]. Moreover, TGFβ and TCR activation together lead to “inside-out”-signaling of CD103 [35,36]. In parallel, the stimulation of both TCR and chemokine receptors prompts conformational changes in CD103 and thus an increased affinity to E-cadherin [53,54,55]. In turn, CD103 leads to activation of the paxillin pathway through close interaction with its ligand, E-cadherin. The binding of paxillin to the cytoplasmic domain of CD103 facilitates the migration and effector functions of cytotoxic T cells. This results in cytokine secretion (interferon (IFN)-γ, tumor necrosis factor (TNF), T cell intracellular antigen (TIA)-1, granzyme), re-localization of cytotoxic granules and their release (under TCR co-stimulation), as well as TCR-mediated cytotoxicity and, eventually, the demise of the target cell (Figure 2) [56].

On the tissue level, CD103 promotes the infiltration of T cells into human E-cadherin-expressing tumor islets in vitro and contributes to early intratumoral T cell signaling [51]. It thus facilitates a strong connection and cytotoxicity between the CD103^+^CD8^+^ T_RM_ and the epithelial target cell. Knocking down E-cadherin prevented this close interaction. The binding of CD103 to E-cadherin also affords cytokine secretion and morphological stabilization of the so-called cytotoxic immunological synapse. Together, these CD103-mediated activities are thought to contribute to effective tumor cell killing [16,35,41,57]. Indeed, the potential of cytotoxic T lymphocytes (CTL) to kill certain E-cadherin-bearing tumor cells correlated directly with their CD103 expression in some cases [35]. Of note, CD103^+^ CTL express the active form of TGF, thus continually self-regulating CD103 expression in an autocrine fashion without relying on external TGF supply. These cells also show improved TCR antigen sensitivity, which enables faster cancer recognition and rapid antitumor cytotoxicity. In addition, they migrate faster, have increased co-expression of inhibitory receptors, and may undergo apoptosis following prolonged exposure to cancer cells [58].

It is not yet entirely clear whether CD103 primarily facilitates the lysis of tumor cells through epithelial retention of CTLs or contributes directly to the killing of E-cadherin-bearing cancer cells. In any case, CD103-expressing CTLs were able to lyse fibroblasts after the latter were transfected with E-cadherin [59]. Likewise, there appears to be a direct correlation between CD103 expression by human CTLs and their capacity to kill E-cadherin-bearing cancer cells [35]. Inhibition of CD103 and downregulating E-cadherin inhibited tumor cell killing [15].

For effective cytolysis, CTLs form a ring-shaped synapse-like structure with their target cells. These structures, also called cytotoxic immunological synapse or pSMAC (peripheral supramolecular activation cluster), are built primarily by binding of LFA-1 to ICAM-1 [60]. CD103/E-cadherin can substitute for pSMAC generation when ICAM-1/LFA-1 are absent [16,41]. Even when LFA-1/ICAM-1 are present, CD103 helps to attract cytolytic granules to the synapse in a PLCγ and F-actin-dependent manner and contributes to the recruitment of CCR5 (C-C-motif-chemokine-receptor-5, CD195) to the synapse [41,61]. Since TGFβ induces CD103 and suppresses LFA-1 within the tumor microenvironment [51], it is quite conceivable that, at least in some epithelial tumors, cytolysis by CTL depends on the interaction between CD103 and E-cadherin within the synapses.

To add another layer of complexity, there are also intra- and peritumoral regulatory T cells expressing CD103 (CD103^+^/CD25^+^ T_reg_) that may even enhance tumor growth [62]. This activity also appears to be related to CCR5 recruited by CD103, so that overall, there is a finely regulated feedback system whose net contribution to tumor biology is difficult to predict on a case-by-case basis. Gastric carcinoma is a very instructive example of seemingly opposite effects of CD103-expressing T cells: while the infiltration of cytotoxic CD8^+^CD103^+^ T cells is associated with a better prognosis for patients [63], the exact opposite has been reported for infiltration with CD4^+^CD103^+^ T cells, likely owing to immunoevasion [64].

The role of CD103 in the pathobiology of solid tumors is arguably even more complex and extends beyond its function in cytotoxic T cells. A robust body of research has focused on, for example, CD103-expressing macrophages or mast cells in the context of tumors. CD103 is also found on tumor-associated dendritic cells (DC) [65], and expansion of such cells may augment the response to immunological and targeted therapies [66]. Indeed, vaccination with CD103^+^ DC appeared to limit tumor growth of both primary and metastatic tumors and also resulted in a higher response rate to immune checkpoint blockade (ICB) in a murine model [67]. In addition, chemokines missing within the tumor microenvironment, especially CCL4, could impair the functionality of CD103^+^ DC, which in turn can diminish the efficacy of ICB. When CCL4 was introduced into murine tumors through a collagen-binding domain, CD103^+^ DCs were recruited and the response to ICB was improved [68]. Under experimental conditions, antitumoral therapy with adenosine receptor antagonists enhanced antigen presentation by CD103^+^ DC [69]. Recent work suggests that CD103-expressing exosomes derived from tumor stem cells through miR that impair PTEN expression may promote epithelial–mesenchymal transition and metastasis of renal cancer cells and that these exosomes have a degree of organotropism [70].

## 4. Prognostic Implications of CD103 Expression in Tumor-Infiltrating Immunocytes

Given the need for reliable biomarkers predicting the course and response to therapies of malignant tumors, CD103 has moved into the limelight of both clinicians and researchers’ scientific interests. Its putative role—emerging from a growing number of studies—suggests that it may help to predict the disease course and response to therapies in some cases. Yet, the picture is not uniform, and many details still remain to be elucidated.

The presence of many tumor-infiltrating lymphocytes (TIL) is thought to reflect an active anti-tumoral immune response. In such a case, the respective tumor is referred to as “hot”. Indeed, in several solid tumor entities, it appears that the number and density of TIL correlate with disease-specific survival [71,72,73]. Consequently, the lymphocytic infiltrate itself is a prognostic marker in these cases. However, various specific traits of the infiltrate, especially the composition of its T cell subsets, also appear to influence both the course and the response to therapy. Distribution and expression of CD103 by TIL within the tumor tissue itself or in the peritumoral stroma have been investigated in several epithelial tumor entities. Alas, the comparability of available data is limited, as lymphocyte subpopulations were examined with different methods and in different clinical settings. However, in many cases, the presence of CD103^+^CD8^+^ T cells within the tumor and/or its stroma seems to be associated with a more favorable prognosis (Table 1).

CD103^+^ TILs were detected in non-small cell lung cancer (NSCLC) within both the stroma and the tumor tissue itself, albeit at smaller amounts within the latter compartment [15]. CD103 appeared to promote T cell infiltration into the epithelial compartment of the tumor, and stage 1 patients with CD103^+^ TIL within the tumor epithelium showed an improved overall survival [15]. Of note, the density of CD103-expressing TIL varies widely between individual patients. Transcriptomic profiling of CTL showed that high densities of CD103^+^ TIL correlated with increased cytotoxicity and higher antitumor immunity. In lung tumors, some of these cells have been identified as Tc17 cells (cytotoxic T cells producing IL-17) whose numbers are increased in responders to anti-PD-1 therapy [90]. Their activity is presumably effected through the release of cytotoxic molecules such as granzyme, perforin, lysosomal-associated membrane protein 1 (LAMP-1, CD107a), interferons, lymphocyte-activation gene-3 (LAG-3, CD223), T cell immunoglobulin and mucin domain 1 protein (TIM), programmed death-ligand 1 (PD-L1, CD274), and cytotoxic T lymphocyte-associated protein 4 (CTLA-4, CD152) [74]. When patients with NSCLC and pulmonary squamous cell carcinoma (pSCC) were studied, 10–30% of the CD8^+^ TILs in NSCLC were positive for CD103. In pSCC, higher expression of CD103 by intratumoral CD8^+^ T cells compared to stromal cells was associated with increased expression of PD-1-associated molecules and a better prognosis for the patients. It was hypothesized that the interaction of CD103 with E-cadherin facilitates the polarization of cytotoxic granules and exocytosis [75], and CD103^+^CD8^+^ cells seem to respond preferentially to PD-1 blockade [76].

Likewise, a high number of CD103^+^ TILs correlated with improved survival and response to therapy in patients with breast, ovarian, endometrial, or cervical cancer [80,81,82,83,85]. Interestingly, prior vaccination against human papillomaviruses in patients with cervical cancer led to an increase of CD103^+^CD8^+^ TIL by 25% and, consequently, to a better prognosis in terms of survival and response to therapy [84]. There is also an association with PD-1 expression in ovarian cancer [83].

In urothelial (transitional cell) carcinoma of the bladder, the number of intratumoral CD103^+^CD8^+^ TILs correlated with E-cadherin expression, and the tumor size correlated inversely with the density of these cells. Again, a CD103^+^CD8^+^-rich lymphocytic infiltrate was associated with longer recurrence-free and overall survival of the patients [48].

Infiltration of CD103^+^CD8^+^ TIL in oropharyngeal cancers [87] or cancers of the gastrointestinal tract [63,86,88,89] is also associated with a more favorable prognosis. In gastric cancer, a higher number of CD103^+^CD8^+^ TIL correlates with a better response to adjuvant chemotherapy. In addition, after ICB, the CD103^+^CD8^+^ infiltrate indicates a stronger activation of anti-tumor immune functions compared to a CD103^-^CD8^+^ infiltrate [63].

In cutaneous melanoma, CD8^+^ TIL counts have a higher prognostic value than total CD8^+^ T cell counts. The CD103^+^CD8^+^ subset showed higher expression of PD-1 and LAG 3, both of which are relevant for ICB therapy. Furthermore, these cells expanded early after ICB, indicating a response to the treatment. CD103^+^CD8^+^ T cells were also found in micrometastases [78]. In search for regulators, a strong correlation between IL-15 levels and CD103^+^CD8^+^ TIL numbers was found [77]. Loss of E-cadherin resulted in poor response to ICB, while E-cadherin knock-in increased responsiveness [47]. Likewise, CD103-deficient mice [17] lost E-cadherin-dependent tumor control [47].

## 5. CD103 in Epithelial Skin Tumors

The expression of CD103 in epithelial skin tumors has hardly been studied. This is surprising since (i) basal cell carcinomas (BCC) [91,92,93] and cutaneous squamous cell carcinomas (cSCC) [94,95,96] are by far the most common tumors of (especially fair-skinned) humans, (ii) these tumors are usually immunogenic [97,98,99], (iii) they often respond to topical and/or systemic immune modulation [100,101,102,103,104], and (iv) their incidence increases greatly upon immunosuppression [105,106]. Moreover, it was already described more than a quarter of a century ago that malignant T cells of cutaneous T cell lymphomas with high CD103 expression migrate preferentially into the E-cadherin-bearing epidermis [107,108].

Only very recently, the first investigation on CD103 in infiltrating cells of cutaneous squamous cell carcinomas (cSCC) was published. This elegant study showed that cutaneous squamous cell carcinomas have a rather variable number of tumor-infiltrating CD103^+^CD8^+^ cells. These cells expressed more IL-10, PD-1 and CTLA-4 compared to CD103-negative cells. Moreover, the number of CD8^+^CD103^+^ cells seemed to correlate with a rather unfavorable clinical course, which seems to contrast quite strikingly with findings in solid tumors of several other organs [79]. However, CD103 distribution in other epithelial skin tumors such as basal cell carcinomas or benign neoplasms is unknown. In addition, data on the spatial distribution of CD103-expressing cells in epithelial tumors (epithelial tumor fraction vs. tumor stroma) including cSCC are not available.

Therefore, we performed a study on the expression of CD103 in basal cell carcinomas and squamous cell carcinomas of the skin following approval by the ethics committee of the University Medical Center Göttingen (no. 19/3/17).

To this end, histologic specimens of 39 BCC (including 15 nodular, 10 superficial, 9 morphaeaform, 3 mixed nodular/morphaeaform, and 2 fibroepithelioma of Pinkus type), as well as 22 cSCC (15 highly differentiated including keratoacanthomas, 6 poorly differentiated including Bowen’s carcinoma, one tumor not further classified) were examined. For comparison, we studied 5 Merkel cell carcinomas (non-keratinocytic epithelial skin cancer) and 6 seborrheic keratoses (benign keratinocytic neoplasia). Paraffine-embedded sections of all tumors were stained with hematoxylin and eosin as well as immunohistochemically with antibodies against CD103, E-cadherin, and N-cadherin. Analysis of CD103 expression in all tumors was semiquantitative using a relative five-point scale ranging from 0 (no expression) to 4 (very dense infiltration of CD103-expressing cells). Tumor nests (epithelial tumor tissue in the strict sense) and peri- and intratumoral stroma (mesenchymal areas) were evaluated separately. Expression of E- and N-cadherin was also evaluated semiquantitatively and correlated with the presence of CD103-expressing cells.

In general, CD103 expression in epithelial skin tumors was considerably heterogeneous within and between tumors (Figure 3a), ranging from (almost) no CD103-positive cells (score 0) to within tumor nests and/or peritumoral stroma very dense infiltrate of CD103-positive cells (score 4), with strong expression occurring exclusively in some cSCC. Direct comparison of cSCC and BCC revealed an overall significantly higher expression density in cSCC (mean score 2.19, SEM 0.21) compared to BCC (mean score 0.59, SEM = 0.14; *p* < 0.00001) (Figure 3b). Both seborrheic keratoses and Merkel cell carcinomas also showed clearly lower expression of CD103 compared to cSCC. A clear assignment to specific histological growth types or differentiation patterns of BCC and cSCC could not be established, although more de-differentiated cSCC or Bowen’s carcinomas showed a (statistically non-significant) tendency towards a higher density of CD103-positive cells. The significantly higher density of CD103-expressing cells in cSCC compared with BCC was detectable in both tumor tissue and stroma (Figure 3c). Surprisingly, however, the density of CD103-positive cells in tumor and stroma did not correlate strongly within both the cSCC and the BCC cohorts (correlation coefficients 0.756 in cSCC and 0.534 in BCC, respectively). Figure 4 shows examples of the distribution of CD103-expressing cells in cSCC (Figure 4a), BCC (Figure 4b), and SK (Figure 4c). There is no overt association of the infiltration of CD103-expressing cells with the presence of E-cadherin in the respective epithelial skin tumors. These observations suggest that infiltration of CD103-expressing cells into the tumor and stromal tissue of epithelial skin tumors may be regulated by at least partially different mechanisms.

## 6. Critical Appraisal of our Current Knowledge and Some Areas of Future Research Needs

We still do not know many details of how CD103 interacts with tumor cells or peritumoral stroma. Similarly, it is still largely unclear how exactly CD103 affects or is affected by adaptive or innate immune processes. This also applies to the relevance of CD103 in therapy responses. These statements hold true notwithstanding that translational scientific evidence is descriptively uncovering associations between CD103 expression and the behavior of tumors. However, several starting points for future research are emerging.

In several cases, CD103 expression is associated with increased expression of molecules that are important for modern oncological checkpoint inhibitors. However, this association is so far only descriptive; it is still unclear whether and what regulatory and/or functional relationships exist. Individual responses to ICB vary greatly [109]. It seems that factors such as density, distribution (compartmentalization), and precise composition of different sub-populations of immune cells may critically influence the effect of ICB [110].

In several solid tumors, the response to immunotherapy was improved when CD8^+^CD103^+^ cells were present in the infiltrate. Given that a stronger immune cell infiltrate in tumors may be associated with higher susceptibility to immunotherapies, it seems reasonable to promote survival and functionality of cytotoxic T cells in and around tumors. For example, the function of CD103^+^CD8^+^ T cells is restored after PD-1 inhibition in gastric cancer [63]. Along this line, peripheral blood CD8^+^ cells rapidly upregulated CD103 after activation by HGSC (high-grade serous carcinoma) cells under TGF-rich conditions [84]. This led to the idea that CD103^+^ TIL arise in response to an adaptive immune response. Consequently, T cell activation markers such as PD-1 and CD137 (tumor necrosis factor receptor superfamily member 9, TNFRSF9) are also elevated on CD103^+^ TIL [111]. Checkpoint inhibitors may reactivate CD103^+^ TIL and thus lead to cancer eradication. Consequently, targeted induction of CD103^+^ cells could be a worthwhile approach.

In light of the above, future investigations also should reasonably further clarify the potential of CD103^+^ TIL in anti-cancer immunotherapies. Such approaches could be combined with targeted cellular therapies (e.g., CAR-T cells) or vaccination. In addition to immuno-oncological therapy, such as PD-1 blockade, it may also be worthwhile to modulate the composition of TIL subsets, e.g., by means of CAR technology, which can be modified to include the expression of other relevant molecules [112] such as CD103. Similarly, CD103^+^ T cells could enhance the antitumor response in the presence of concomitant vaccination, although this remains somewhat speculative at present. Along this line, co-expression of CD103 and CD39 appears to identify some tumor-infiltrating CD8^+^ TIL specifically induced in the tumor microenvironment. A higher density of these cells in human head and neck cancer correlated with improved overall survival. These cells had the ability to kill autologous tumor cells [49], and it has been proposed that they indicate responsiveness to treatment and that they can be adoptively transferred to treat a tumor [113]. However, there are no clinical studies yet to support this hypothesis.

In this context, it would also be relevant to further elucidate the role of locally applied or produced TGF, the main stimulator of CD103 expression. Moreover, it seems important to further explore the apparently dichotomous role of CD103 in some cases, i.e., its association with pro- versus antitumor effects.

Although infiltration of CD103-positive cells appears to correlate with the expression of E-cadherin in some tumors, such as urothelial carcinoma [48] or melanoma [47], this is apparently not always the case. Epithelial skin tumors, particularly cSCC and BCC which we have highlighted here, are a conspicuous example of this notion. This raises the question, on the one hand, whether and to what extent CD103-expressing T cells migrate into E-cadherin-negative tumors and, on the other hand, whether and what functions CD103 might exert at the molecular and cellular levels if it cannot bind to its major ligand.

Regarding the first question, at least two possible answers can be delineated which are not mutually exclusive: (i) CD103-expressing cells could use other adhesion receptors to migrate into tumors. In this case, CD103 expression would not be required for the process of immigration and retention. In fact, T cells express several other receptors, such as β1 and β2 integrins, which can bind to immunoglobulin superfamily ligands (e.g., ICAM-1) on various tumor cells or other cells to form immuno-oncologic synapses. In these, CD103 could then exert important regulatory or effector functions. In this context, however, it is still unclear whether CD103 might transmit different signals depending on ligand binding. Conformational changes in CD103 at least seem to significantly influence cellular motility and interaction with E-cadherin-expressing epithelial cells [55]. (ii) It is conceivable that CD103 binds to ligands other than E-cadherin. In this context, CD103 may bind to as yet unidentified ligands on endothelia [30] or certain epithelia [29]. In addition, recent reports suggest that cadherin 26 may be a ligand for CD103. Evidence for the latter was found in lung tissue [31]. Whether and to what extent this putative function plays a role in tumors cannot yet be verified due to the lack of suitable detection reagents.

Regarding the second question, there is only very little scientific evidence related to immunologic interactions with tumors. While for CD4^+^ T cells it seems clear that CD103 is associated with regulatory functions even in the absence of binding to E-cadherin [9,10], such evidence is not yet available for CD8^+^ T cells. However, based on clinical and histological correlation studies [15,63,80,86], it is reasonable to assume that in some tumors CD103-expressing TILs enhance antitumor immune responses even in the absence of E-cadherin expression. However, that this is not true for all tumors is suggested by our observations and a recent study [79] in cutaneous squamous cell carcinoma.

In addition to the functional relevance of CD103-expressing cells, their utility as prognostic biomarkers is an important issue. Indeed, it is emerging that CD103^+^ lymphocytes in and around tumors seem to correlate with certain courses of the diseases. However, the picture is by no means uniform and there is limited comparability between the available data. Although it would be desirable from a clinical-practical point of view and is propagated accordingly in some publications, we currently have to conclude that the expression or absence of CD103 in tumors alone is not (yet) a sufficient prognostic biomarker.

One factor that may prohibitively complicate the assessment of prognostic significance is the very heterogeneous expression of CD103 within the same tumor entity. Although in several tumors statistical evidence could be obtained so that, on average, increased infiltration of CD103-expressing TIL might be related to a rather favorable prognosis, these findings—as was also seen with other hopeful biomarkers—do not (yet) allow reliable conclusions to be drawn for the respective individual case. Furthermore, the density of CD103-positive cells even within a tumor (or its metastases) can vary considerably, so that pathobiological differences in different regions of the tumor could also be relevant here. Based on the findings available so far, it is possible that “microcompartments” exist within tumors in which local interactions with the immune system are modulated by CD103. To what extent this affects the behavior of the entire tumor can only be speculated at present. If the hypothesis that CD103-expressing TILs exert stronger antitumor activity in the corresponding tumor regions is correct, it is conceivable that regions with no or low presence of CD103^+^ cells gain a relative growth advantage and immune-evasive selection benefit.

## 7. Conclusions

CD103 remains an exciting molecule especially in the context of antitumor immune responses. It seems to play a role in the orchestra of complex pro- and antitumoral players. However, this role is by no means completely understood—both in terms of biomarker properties and functional involvement in tumor-directed immune responses. However, we are convinced that further research in this direction is warranted and important.

## Figures and Tables

**Figure 1 cancers-13-06211-f001:**
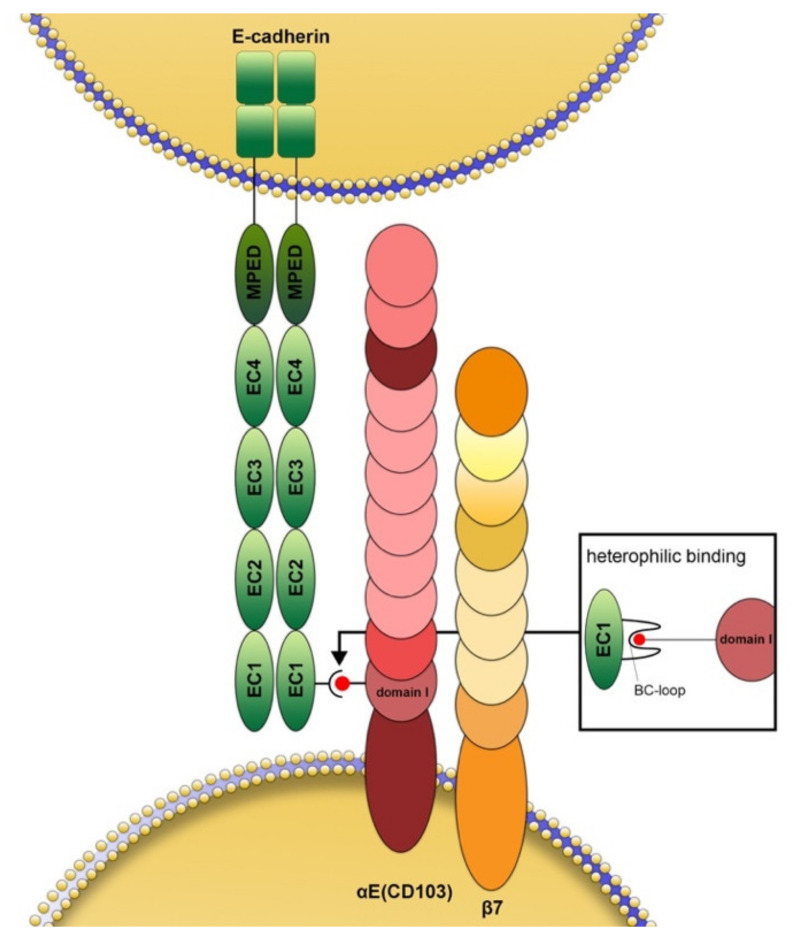
Schematic structure and interaction of CD103 with E-cadherin. The αE chain (CD103 in the stricter sense) is extracellularly composed of an X (extra) domain, which only occurs in this integrin chain and contains a proteolytic cleavage site, an I (inserted) domain as well as a “thigh”, two “calf” and a propeller domain. In heterodimeric association with the integrin β7 chain (CD103 by extension), the βE chain binds to the tip of the so-called BC loop (small box) of the extracellular EC1 domain of E-cadherin with the help of its MIDAS motif of the first extracellular domain. This binding site is different from the one responsible for the homophilic binding of E-cadherin. The β7 chain does not bind directly to E-cadherin. It facilitates intracellular signaling through its tail domain. This subunit features 4 EGF-like domains, 2 so-called hybrid domains separated by the β-I-domain, and a plexin/semaphorin/integrin domain.

**Figure 2 cancers-13-06211-f002:**
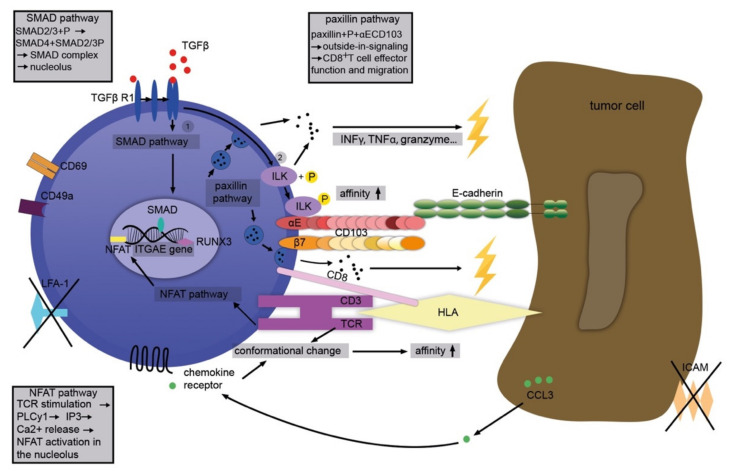
Complex immunological functions associated with CD103. TGFβ stimulates the SMAD pathway, transcriptionally upregulating CD103. At the same time, TGFβ also increases the affinity of CD103 for its ligand, E-cadherin, through phosphorylation of integrin-linked kinase (ILK). Enhanced binding affinity via conformational changes in CD103 is also achieved through activation of chemokine receptors (for example, by CCL3). TCR signaling also enhances CD103 expression via NFAT. Together, these elements of “inside out” signaling culminate in the facilitation of ligand binding. Binding of CD103 to E-cadherin, on the other hand, causes or enhances various effector functions in T cells as elements of “outside-in” signaling. These include, for example, activation of the paxillin signaling pathway with effects on cell migration, cytokine secretion, re-localization, and depletion of cytotoxic granules, and ultimately cytotoxicity and death of the target cell.

**Figure 3 cancers-13-06211-f003:**
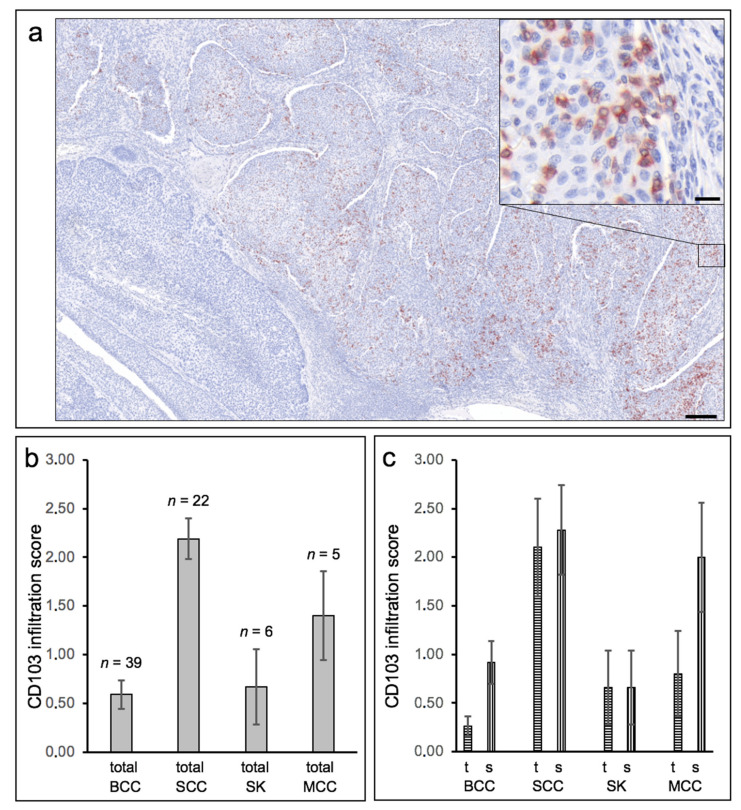
Heterogeneous distribution of CD103-expressing cells and preferential expression in cutaneous squamous cell carcinomas. (**a**) Low-power overview image of a histological specimen of a cutaneous squamous cell carcinoma. Immunohistochemical staining was performed against CD103 (clone EP206, CellMarqe/Medac, Wedel, Germany). The very heterogeneous intratumoral distribution can be seen in this example. In the lower left part of the tumor there are no CD103-positive cells at all, while the remaining part of the tumor shows a dense infiltration of these cells. Length bar 200 µm. The part of the tumor marked on the right has been further enlarged. The inserted high-power box in the upper right corner shows the localization of CD103-expressing lymphocytes between the tumor cells (**left** part) and also (somewhat weaker here) within the stroma (**right** part). Length bar = 20 µm. (**b**) The density of infiltrating CD103-expressing cells was determined semiquantitatively in basal cell carcinomas (BCC), squamous cell carcinomas (SCC), seborrheic keratoses (SK), and Merkel cell carcinomas (MCC) using a five-point scale (0–4). SCC show the strongest infiltration of CD103-expressing cells on average (*p* < 0.00001 compared to BCC). (**c**) Semiquantitative evaluation was also performed separately for infiltration of CD103-expressing cells directly in epithelial tumor tissue (t) and in peritumoral stroma (s). Here, too, a predominance in SCC is evident, although within individual tumors, the CD103 density in tumor tissue and stroma do not correlate well.

**Figure 4 cancers-13-06211-f004:**
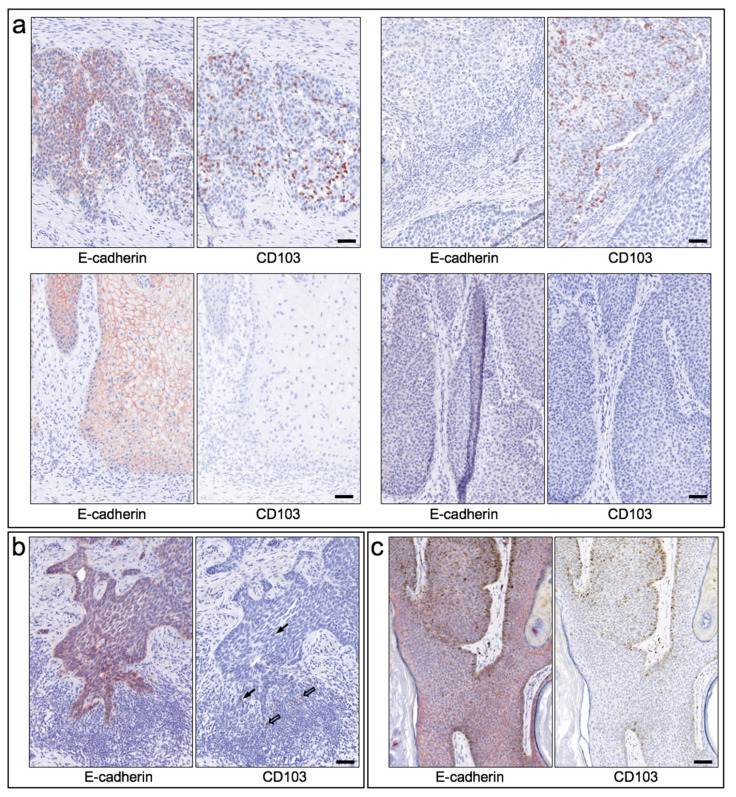
Infiltration of CD103-positive lymphocytes in epithelial skin tumors and expression of the ligand E-cadherin. (**a**) Exemplary immunohistochemical staining of consecutive cryostat-cut sections of four SCC is shown, staining for E-cadherin on the left and staining for CD103 on the **right**, respectively. Top left: Clear expression of E-cadherin in the tumor and infiltration of numerous CD103-expressing cells in the corresponding tumor tissue. Top right: Absence of E-cadherin expression and nevertheless infiltration of numerous CD103-positive cells. Bottom left: strong expression of E-cadherin and lack of immigration of CD103-expressing cells. **Bottom right**: Neither E-cadherin nor CD103 are expressed appreciably. Scale bar = 50 µm. (**b**) Exemplary staining of a BCC for E-cadherin (**left**) and CD103 (**right**). Despite clear E-cadherin expression, very few CD103-positive cells migrate into the tumor tissue (black arrows). In the dense peritumoral lymphocytic infiltrate, also only few cells express CD103 (open arrows). Scale bar = 50 µm. (**c**) Exemplary staining of an SK for E-cadherin (left and CD103 (**right**). E-cadherin is clearly expressed, CD103-positive cells are not detectable (in this example). Scale bar = 50 µm.

**Table 1 cancers-13-06211-t001:** CD103^+^ CTL and correlation with prognosis in different types of epithelial tumors. The table lists the epithelial tumor types in which the role of CD103^+^ CTL has essentially been studied and correlated with clinical courses to date. The cell types and mediators investigated and the main findings are summarized. A direct comparison is only possible to a very limited extent, as the parameters investigated vary considerably between the different studies.

Tumor Entity	Phenotype Studied	Prognostic/Therapeutic Value	Reference
Non small cell lung cancer (NSCLC)	CD103^+^CD8^+^ TIL PD-1^+^ TIM^+^ Granzyme B^+^	Improved survival of patients in early-stage disease; increased response to PD1-blockade; improved disease-free survival of patients.	[15,74,75,76]
Pulmonary SCC (pSCC)	CD103^+^CD8^+^ iTIL CD103^+^ TIL	Improved disease-free survival and overall survival of patients.	[75]
Melanoma	CD103^+^CD69^+^CD8^+^ TIL PD-1^+^ LAG3 IL-15	Association of CD103^+^CD69^+^CD8^+^ TIL with improved melanoma-specific survival in treatment-naïve patients; expansion of this subset under ICB indicates response to treatment.	[77]
	CD103^+^	E-cadherin expression enhances CD103 antimelanoma responses; E-cadherin loss downregulates response to ICB in patients; E-cadherin knock-in increases responsiveness to ICB in mice; CD103^+^ TIL may support control of mircometastases in humans.	[47]
	CD103^+^CD8^+^	Reduced survival in primary cutaneous melanoma with high *ITGAE* (encoding for CD103).	[78]
Cutaneous SCC (cSCC)	CD103^+^CD8^+^ T_RM_ IFNβ, TNFβ, IL-2 PD-1	Higher expression of CD103 associated with metastasis.	[79]
Breast cancer	CD103^+^CD8^+^ TIL	CD103^+^CD8^+^ density associated with disease-free and overall survival in triple-negative breast-cancer patients; CD103^+^ iTILs predict better prognosis.	[80]
	CD103^+^CD8^+^ TIL	CD103 associated with tumor size, grade and ER/PR status; CD103^+^ iTIL density associated with overall survival and recurrence-free survival in basal-like subtype tumors.	[81]
Ovarian cancer	CD103^+^CD8^+^ TIL CD103^+^CD8^+^PD1^+^	CD103^+^ TILs associated with disease-specific survival in high-grade serous ovarian cancer; better prognosis in patients with CD103^+^CD8^+^ TIL.	[82]
	CD103^+^CD8^+^PD1^+^ TIL TIM-1, CTLA-4, LAG-3	Improved disease-specific survival in high-grade-serous carcinomas; antitumor activity of CD103^+^CD8^+^ cells	[83]
Cervical cancer	CD103^+^CD8^+^ TIL CD137, CTLA4, PD-1, PD-L1	CD103^+^CD8^+^TILs associated with better disease-specific and disease-free survival in patients. Vaccination with sFVeE6/E7 correlates with infiltration of CD8^+^CD103^+^ TIL.	[84]
Endometrical cancer	CD103^+^CD8^+^ TIL PD-1	Number of CD103^+^ cells correlate with prognosis	[85]
Bladder cancer	CD103^+^CD8^+^ TIL	Number of CD103^+^ iTILs associated with overall and recurrence-free survival.	[48]
Esophageal SCC	CD103^+^CD3^+^ TIL	Density of CD103^+^ cells associated with overall and disease-free survival.	[86]
Head neck SCC (HNSCC)	CD103^+^CD8^+^ T_RM_	High number of CD103^+^ cells associated with better overall survival of patients and lower risk of loco-reginal recurrence.	[87]
	CD103^+^CD39^+^CD8^+^ TIL	A higher frequency of CD8^+^CD39^+^CD103^+^ T cells in human tumors correlate with a greater overall survival. CD8^+^CD39^+^CD103^+^ T cells indicate a therapeutic method is effective for treating a tumor.	[49]
Gastric cancer	CD103^+^CD8^+^ TIL	More benefit from adjuvant chemotherapy and better overall-survival in patients with high CD103^+^CD8^+^ TIL numbers; superior anti-tumor effect after ICB	[63]
Colon cancer	CD103^+^CD8^+^ TIL	CD103^+^CD8^+^ TIL associated with better overall survival and inversely correlated with advanced TMN stage and distant metastasis; cancers with microsatellite instability associated with increased numbers of CD103^+^CD8^+^ intraepithelial lymphocytes.	[88,89]

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
