# Peer review of "Integrin αE(CD103)β7 in Epithelial Cancer"

_cancers, 2021, doi:10.3390/cancers13246211_

Round 1

Reviewer 1 Report

The authors should cite the work of Weinberg et al (https://worldwide.espacenet.com/publicationDetails/biblio?DB=EPODOC&II=4&ND=3&adjacent=true&locale=en_EP&FT=D&date=20200514&CC=US&NR=2020149008A1&KC=A1), regarding the use of CD39 and Cd103 for identification of human tumor reactive T cells for treatment of cancer. In this study, the phenotype and function of tumor-reactive CD8 T cells was evaluated, and it was determined that co-expression of CD103 and CD39 identified a population of tumor-infiltrating CD8 T cells specifically induced in the tumor microenvironment. These cells, which are chronically stimulated within the tumor, were highly enriched for tumor reactivity and had the ability to kill autologous tumor cells. A higher frequency of CD8+CD39+CD103+ T cells in human tumors correlated with a greater overall survival. It was determined that the presence of CD8+CD39+CD103+ T cells can indicate that a therapeutic method is effective for treating a tumor. In addition, CD8+CD39+CD103+ T cells can be adoptively transferred into a subject to treat a tumor.

The authors should address in the therapy section the existence, or not, of drugs and / or agonist antibodies against CD103. Likewise, they should address those clinical studies where CD103 is involved. This last, to give support to the last paragraph of the summary where it is indicated that therapeutic aspects are addressed.

Author Response

Dear Editor:

Enclosed I am sending you our revised manuscript in which we have addressed all reviewer comments. The respective changes are highlighted in red font.

Specifically, the following changes have been made:

R1: Comments and Suggestions for Authors

Q1: The authors should cite the work of Weinberg et al (https://worldwide.espacenet.com/publicationDetails/biblio?DB=EPODOC&II=4&ND=3&adjacent=true&locale=en_EP&FT=D&date=20200514&CC=US&NR=2020149008A1&KC=A1), regarding the use of CD39 and Cd103 for identification of human tumor reactive T cells for treatment of cancer. In this study, the phenotype and function of tumor-reactive CD8 T cells was evaluated, and it was determined that co-expression of CD103 and CD39 identified a population of tumor-infiltrating CD8 T cells specifically induced in the tumor microenvironment. These cells, which are chronically stimulated within the tumor, were highly enriched for tumor reactivity and had the ability to kill autologous tumor cells. A higher frequency of CD8+CD39+CD103+ T cells in human tumors correlated with a greater overall survival. It was determined that the presence of CD8+CD39+CD103T cells can indicate that a therapeutic method is effective for treating a tumor. In addition, CD8+CD39+CD103T cells can be adoptively transferred into a subject to treat a tumor.

A1: We have included in our manuscript the patent of A. Weinberg's group proposed by the reviewer and cited it on page 12, paragraph #3. We also explicitly state that the CD8+CD103+CD39+ T cells described by Weinberg's group could indicate increased antitumor activity. In addition, we have also cited the group's underlying original paper in the same place (T. Duhen et al, Nat Commun 9, 2724, 2018). Finally, we added this information to Table 1.

Q2: The authors should address in the therapy section the existence, or not, of drugs and / or agonist antibodies against CD103. Likewise, they should address those clinical studies where CD103 is involved. This last, to give support to the last paragraph of the summary where it is indicated that therapeutic aspects are addressed.

A2: In preparation of this manuscript, we have of course intensively searched for clinical studies or therapeutic reagents related to CD103 expressing T cells in tumor therapy. However, apart from the work already cited in response to Q1 of this reviewer and the corresponding patent, we unfortunately did not find anything.

There is some (mostly preclinical) work in which blockade of CD103 has been used in the context of transplant rejection (e. g., Xue D, et al. Cell Death Dis. (2019); 10(10):735. or Zhang and Hadley. Chin Med J (Engl). (2010); 123(24):3644-51. or Zhang et al. Am J Transplant. (2009); 9(9):2012-23.). In addition, the importance of CD103 on cells of the innate immune system such as dendritic cells in clinical contexts is also being investigated (e. g., Yuan et al. Transplant Proc. (2013);45(9):3408-13.). However, these aspects are beyond the scope and objective of our work. We have therefore refrained from extending our manuscript in this direction and we hope for the agreement of the reviewers and the editors. This response also refers to Q2 of reviewer #3.

Reviewer 2 Report

Johanna C. Hoffmann and  Michael P. Schön has enlighten the role of  Integrin αE(CD103)β7 in epithelial cancer . They have very nicely shown the αE(CD103)β7 integrin expressing tumor infiltrating T  lymphocytes in epithelial tumors. This receptor enhances the tumor-infiltrating immune cells through interaction with its ligand, E-cadherin. They have nicely explained the regulatory functions of  αE(CD103)β7 and showed that TGF-rich environments enhances the binding strength between tumor cells and infiltrating T lymphocytes. Authors have also described that αE(CD103)β7 is barely present in basal cell carcinomas, but much more abundant in squamous cell carcinomas with heterogeneous distribution. This review is very important to understand the role of αE(CD103)β7 and could be a valuable review to understand the role of αE(CD103)β7 in pathophysiology, therapy and prognosis of solid tumors. Overall, I am completely satisfied with this review, however, authors should refine Abstract and give a proper message in Abstract.

Author Response

R2: Comments and Suggestions for Authors

Q1: Johanna C. Hoffmann and  Michael P. Schön has enlighten the role of  Integrin αE(CD103)β7 in epithelial cancer . They have very nicely shown the αE(CD103)β7 integrin expressing tumor infiltrating T  lymphocytes in epithelial tumors. This receptor enhances the tumor-infiltrating immune cells through interaction with its ligand, E-cadherin. They have nicely explained the regulatory functions of  αE(CD103)β7 and showed that TGF-rich environments enhances the binding strength between tumor cells and infiltrating T lymphocytes. Authors have also described that αE(CD103)β7 is barely present in basal cell carcinomas, but much more abundant in squamous cell carcinomas with heterogeneous distribution. This review is very important to understand the role of αE(CD103)β7 and could be a valuable review to understand the role of αE(CD103)β7 in pathophysiology, therapy and prognosis of solid tumors. Overall, I am completely satisfied with this review, however, authors should refine Abstract and give a proper message in Abstract.

A1: We thank the reviewer very much for the positive evaluation. There were no specific points to address.

Reviewer 3 Report

Some comments:

  • Table 1 is crowded. Please simplify it, remove written parts in order to make it more schematic;
  • I think a part regarding antibodies and early clinical trials on anti-CD103 is missing. I would consider a different paragraph with a more clinical part with early trials and ongoing studies.

The review is complete, well-written and focusing mainly on the molecular aspects of CD103 in epithelial cancer. Figures are really well-done and informative, I would improve table 1. I would add a more clinical part, considering also early-phase and ongoing trials, if any.

Some comments:

- Table 1 is crowded. Please simplify it, remove written parts in order to make it more schematic;
- CD103+ and epithelial tumor cells: a multifaceted relationship: this paragraph is really too long. Synthesis is necessary, please shorten it.
- I think a part regarding antibodies and early clinical trials on anti-CD103 is missing. I would consider a different paragraph with a more clinical part with early trials and ongoing studies. Perhaps a table in order to list the ongoing studies would be helpful.

Author Response

R3: Comments and Suggestions for Authors

The review is of interest. Some comments:

Q1: Table 1 is crowded. Please simplify it, remove written parts in order to make it more schematic;

A1: The table has been substantially "slimmed down". We have removed detailed information on cellular and molecular interactions in almost every row, and we have condensed the information on diagnostic and prognostic value.

Q2: I think a part regarding antibodies and early clinical trials on anti-CD103 is missing. I would consider a different paragraph with a more clinical part with early trials and ongoing studies.

A2: As outlined in response to reviewer #1 (Q2), we have searched for clinical studies or therapeutic reagents related to CD103 expressing T cells in tumor therapy. However, apart from the work already cited in response to Q1 of reviewer #1 and the corresponding patent, we unfortunately did not find anything. There is some preclinical work on the blockade of CD103 in transplant rejection or regarding the role of CD103 on cells of the innate immune system. There are no trials or studies regarding CD103+ TIL. We have refrained from extending our manuscript in the direction of inflammatory conditions as this would clearly be beyond the scope of this paper, and we hope for the agreement of the reviewers and the editors. This response also refers to Q2 of reviewer #1.

Reviewer 4 Report

This review provides an overview of the role of integrin CD103 in T cell function and anti-tumour activity.  This review was an interesting and informative piece of work, and was generally well-written. However, some statements lacked appropriate references or were incompletely described and explored, and the review contained some minor grammatical errors. These are listed in detail below.

The review component of this piece was relatively brief and was not exhaustive.  Although the description of the role of CD103 in engaging E-cadherin+ targets cells and in promoting immunological synapse formation was detailed, a complete summary of TRM priming, differentiation and overall phenotype was lacking, and the discussion of the role and interactions of CD103+ T cells in the tumour microenvironment and in tumour therapy could have been more detailed.  Further, the text in the final section of the review was somewhat repetitive of the first 6 pages.

The inclusion of the author’s own primary data (although interesting) in the context of a review article was confusing -have the authors considered submitting this study as a primary manuscript instead? I leave it to the editorial staff to make a decision as to the appropriateness of the format of this review for their journal. Altering this may require major revision of the review content and structure. 

General revisions

Page 4: “In turn, CD103 leads to activation of the paxillin-pathway through close interaction with its ligand, E-cadherin. The binding of paxillin to the cytoplasmic domain of CD103 facilitates the migration and effector functions of cytotoxic T cells. This results in cytokine secretion (interferon (IFN) , tumor necrosis factor (TNF), T cell intracellular antigen (TIA)-1, granzyme), re-localization of cytotoxic granules and their release (under TCR co-stimulation), as well as TCR-mediated cytotoxicity and, eventually, the demise of the target cell (Figure 2).” - the authors might consider including https://doi.org/10.1158/0008-5472.can-17-1487 which seems relevant to this section.  

Page 6: The authors state “Recent work even suggests that CD103-expressing exosomes derived from tumor stem cells may promote epithelial-mesenchymal transition and metastasis of renal cancer cells and that these exosomes have a degree of organotropism [70].” but it should be made more clear that while CD103 targets the exosomes to E-Cad+ cells the biological effects in this paper were mediated by miR that impair PTEN expression.

Page 7: Formatting and alignment of table 1 might need to be adjusted.

Page 10: Figure 3 – the snapshot (?) used of the graphs still has the red underlines on some words and will need to be replaced.  The data as shown on the graphs is lacking any indication of statistical significance?

Page 11: Figure 4 - the snapshot (?) used of the graphs still has the red underlines on some words and will need to be replaced. 

Page 12: “Like notions apply to the weighting of specific immune reactions and therapy

responsiveness. In fact, there is already quite a substantial and growing body of data on

the latter aspects. However, it is the molecular and cellular correlations that are important

for the rational development of selective therapies. Having said this, there are nevertheless

quite a few aspects that are now condensing into a mosaic and from which possible

starting points and also needs for future research are emerging.” This section of text in ~confusing and unclear, and should be re-written.

Pages 11-13:  The content of this section seems to be highly duplicative of the content of pages 3-6. 

 Typographical and other revisions.

Page 2: “TGF-ß induces expression of CD103 on CD8+ cells much stronger as compared to CD4+ cells” could be changed to “TGF-ß induces expression of CD103 more strongly on CD8+ cells than CD4+ cells”

Page 4: “In addition, TGF-ß effects directly an increased affinity of CD103 for E-cadherin” could be changed to: In addition, TGF-ß directly increases the affinity of CD103 for E-cadherin”.

Page 8: “Therefore, to fill this gap in a useful way, we performed an own small study on the

expression of CD103 in basal cell carcinomas” – change “an own” to “our own” or just to “a small study”.   

Author Response

R4: Comments and Suggestions for Authors

Q1: This review provides an overview of the role of integrin CD103 in T cell function and anti-tumour activity.  This review was an interesting and informative piece of work, and was generally well-written. However, some statements lacked appropriate references or were incompletely described and explored, and the review contained some minor grammatical errors. These are listed in detail below.

The review component of this piece was relatively brief and was not exhaustive.  Although the description of the role of CD103 in engaging E-cadherin+ targets cells and in promoting immunological synapse formation was detailed, a complete summary of TRM priming, differentiation and overall phenotype was lacking, and the discussion of the role and interactions of CD103+ T cells in the tumour microenvironment and in tumour therapy could have been more detailed.  Further, the text in the final section of the review was somewhat repetitive of the first 6 pages.

A1: To better balance and complete our presentation in accordance with this reviewer's suggestions, we have somewhat expanded the paragraph on the differentiation of TRM in general (page 2, paragraph #1). However, in doing so, we also had to be careful not to make our paper too long (it already exceeded the usual length) and to keep the focus, so we kept the expansion on TRM rather brief. However, to outline the topic more comprehensively, we have also cited two additional recent review papers (Corgnac et al., 2018 and Gebhardt et al., 2018) and we refer the reader to these papers for more detailed reading.

Additional information on the phenotype of specific CD103-expressing TILs has also been included as explained in response to reviewer #1 (Q1).

Q2:The inclusion of the author’s own primary data (although interesting) in the context of a review article was confusing -have the authors considered submitting this study as a primary manuscript instead? I leave it to the editorial staff to make a decision as to the appropriateness of the format of this review for their journal. Altering this may require major revision of the review content and structure. 

A2: Indeed, it is somewhat unusual to combine original data and a review, although there is some precedence for it and it had been communicated with the editorial office upon their request to submit this article. In this case, we consider our approach to be scientifically quite reasonable, since there are almost no data on expression and distribution of CD103-positive TIL especially for the most common epithelial tumors, namely epithelial skin tumors. However, we strongly agree with the reviewer that the presentation of original data should be more clearly communicated and reflected in the manuscript category. We therefore propose to publish our work in the "article" category and we have changed this in the manuscript accordingly.

General revisions 

Q: Page 4: “In turn, CD103 leads to activation of the paxillin-pathway through close interaction with its ligand, E-cadherin. The binding of paxillin to the cytoplasmic domain of CD103 facilitates the migration and effector functions of cytotoxic T cells. This results in cytokine secretion (interferon (IFN) g, tumor necrosis factor (TNF), T cell intracellular antigen (TIA)-1, granzyme), re-localization of cytotoxic granules and their release (under TCR co-stimulation), as well as TCR-mediated cytotoxicity and, eventually, the demise of the target cell (Figure 2).” - the authors might consider including https://doi.org/10.1158/0008-5472.can-17-1487 which seems relevant to this section.

A: We added this article to the review (Gauthier et al., 2017) This issue is also addressed in figure 2.

Q: Page 6: The authors state “Recent work even suggests that CD103-expressing exosomes derived from tumor stem cells may promote epithelial-mesenchymal transition and metastasis of renal cancer cells and that these exosomes have a degree of organotropism [70].” but it should be made more clear that while CD103 targets the exosomes to E-Cad+ cells the biological effects in this paper were mediated by miR that impair PTEN expression.

A:  The information has been added on p.6, paragraph #2.

Q: Page 7: Formatting and alignment of table 1 might need to be adjusted.

A: We simplified the contents of the table quite substantially (also in response to reviewer #3, Q1). Formatting and alignment will be addressed in the production process by the publisher.

Q: Page 10: Figure 3 – the snapshot (?) used of the graphs still has the red underlines on some words and will need to be replaced.  The data as shown on the graphs is lacking any indication of statistical significance?

A: The figure has been corrected and the red lines are deleted.

Q: Page 11: Figure 4 - the snapshot (?) used of the graphs still has the red underlines on some words and will need to be replaced. 

A: The figure has been corrected and the red lines are deleted.

Q: Page 12: “Like notions apply to the weighting of specific immune reactions and therapy

responsiveness. In fact, there is already quite a substantial and growing body of data on

the latter aspects. However, it is the molecular and cellular correlations that are important

for the rational development of selective therapies. Having said this, there are nevertheless

quite a few aspects that are now condensing into a mosaic and from which possible

starting points and also needs for future research are emerging.” This section of text in ~confusing and unclear, and should be re-written.

A: The section has been re-written and shortened considerably (p. 12, paragraph #1).

Q: Pages 11-13:  The content of this section seems to be highly duplicative of the content of pages 3-6. 

A: A certain reminder of some central statements in the chapter on future developments and discussion of current knowledge is arguably helpful with view to the understanding of a readership that does not work exactly in this field. Of course, it should be limited to the necessary extent. This is certainly also a matter of personal taste and the other three reviewers did not comment on this point. We also asked an independent colleague from our institution to read the manuscript with regard to unnecessary redundancy and we did not receive any feedback from him on conspicuities in this respect either. However, in order to comply with this reviewer's suggestion, we have shortened the text in some places on pages 12 to 14. Since this was done in several places, these were not marked individually for the sake of clarity.

Typographical and other revisions. 

Q: Page 2: “TGF-ß induces expression of CD103 on CD8+ cells much stronger as compared to CD4+ cells” could be changed to “TGF-ß induces expression of CD103 more strongly on CD8+ cells than CD4+ cells”.

A: Thank you for this comment. The change has been implemented in the text (p. 3, paragraph #1).

Q: Page 4: “In addition, TGF-ß effects directly an increased affinity of CD103 for E-cadherin” could be changed to: In addition, TGF-ß directly increases the affinity of CD103 for E-cadherin”.

A: Thank you for this comment. The change has been implemented in the text (p. 4, paragraph #2).

Q: Page 8: “Therefore, to fill this gap in a useful way, we performed an own small study on the expression of CD103 in basal cell carcinomas” – change “an own” to “our own” or just to “a small study”.   

A: Thank you for this comment. The change has been implemented in the text (p. 9, paragraph #3).

We hope that we have addressed the comments to your satisfaction and that the manuscript can be accepted for publication in its present form.

Round 2

Reviewer 3 Report

The paper has improved and is suitable for publication

Reviewer 4 Report

Thanks to the authors for their replies and alterations to the manuscript.  As the inclusion of the primary data has been discussed with editorial staff I would recommend the manuscript be accepted in its current form - the only change I would suggest is that as the paper was initially assessed as a review, rather than as a primary article, I believe it would be more appropriate to keep it in the review section. 

Best regards.